# Genome-Wide Identification and Characterization of *Caffeic Acid O-Methyltransferase* Gene Family in Soybean

**DOI:** 10.3390/plants10122816

**Published:** 2021-12-20

**Authors:** Xu Zhang, Bowei Chen, Lishan Wang, Shahid Ali, Yile Guo, Jiaxi Liu, Jiang Wang, Linan Xie, Qingzhu Zhang

**Affiliations:** 1College of Life Science, Northeast Forestry University, Harbin 150040, China; zhangxu2022@163.com (X.Z.); chenbw2021@163.com (B.C.); wongshancrick@126.com (L.W.); Shahidsafi926@gmail.com (S.A.); yileguo@yeah.net (Y.G.); jiaxiliu1122@163.com (J.L.); jiangwang@nefu.edu.cn (J.W.); 2Key Laboratory of Saline-Alkali Vegetation Ecology Restoration, Ministry of Education, College of Life Science, Northeast Forestry University, Harbin 150040, China; 3State Key Laboratory of Tree Genetics and Breeding, Northeast Forestry University, Harbin 150040, China

**Keywords:** *COMT*, gene family, soybean, different stresses

## Abstract

Soybean is one of the most important legumes, providing high-quality protein for humans. The *caffeic acid O-methyltransferase (COMT)* gene has previously been demonstrated to be a critical gene that regulates lignin production in plant cell walls and plays an important function in plant growth and development. However, the *COMT* gene family has not been studied in soybeans. In this study, 55 *COMT* family genes in soybean were identified by phylogenetic analysis and divided into two groups, I and II. The analysis of conserved domains showed that all *GmCOMTs* genes contained Methyltransferase-2 domains. Further prediction of cis-acting elements showed that *GmCOMTs* genes were associated with growth, light, stress, and hormonal responses. Eventually, based on the genomic data of soybean under different stresses, the results showed that the expression of *GmCOMTs* genes was different under different stresses, such as salt and drought stress. This study has identified and characterized the *COMT* gene family in soybean, which provides an important theoretical basis for further research on the biological functions of *COMT* genes and promotes revealing the role of *GmCOMTs* genes under stress resistance.

## 1. Introduction

Soybean is one of the world’s most significant annual leguminous plants, capable of providing humans with high-quality plant protein and oil [1]. As an important economic and food crop, soybean has high nutritional value and a wide range of uses, which profoundly impact the survival and development of mankind [2]. However, soybean lodging and biological or abiotic stress are important factors affecting soybean yield [3]. As a result, increasing soybean lodging and stress resistance is critical for producing high-yielding, stable-yielding, and high-quality cultivars.

A cell wall is a thick wall that exists on the exterior of cells. It plays an important role in the growth and development of plants, such as preserving the shape of cells, enhancing the mechanical strength of cells, and participating in the transmission of information between cells [4]. The cell wall is mainly composed of polysaccharides, proteins, and lignin [5]. Its components enable the plants to respond to biological or abiotic stresses and resist the invasion of pathogens in the first place [6].

Lignin is an indispensable component of cell walls for some higher plants and is the second most abundant biopolymer [7], which can provide mechanical support for plants and facilitate the transportation of water in the whole plant tissue [8]. It is an effective barrier against insects, pathogens, and fungi to plant disease resistance [9], enhancing the lodging resistance of plants [10], which is also involved in plant response to stress [11]. The studies had shown that the mechanical strength of soybean stems was a crucial factor in improving lodging resistance, and lignin had a significant effect on maintaining the mechanical strength of the stems [12]. Wheat cultivars with greater lignin content were also shown to have improved lodging resistance [13]. As a result, lignin plays an essential role in increasing soybean lodging resistance.

The current consensus on the lignin biosynthesis pathway is that it begins with phenylalanine and proceeds via a series of hydroxylation, methylation, ligation, and reduction reactions to produce lignin units [14]. These units are subsequently transferred to the lignification deposition sites, aggregated to form lignin in the secondary cell wall [15]. Lignin is mainly divided into three phenylpropanoid units, including the lignin monomer sinapyl alcohol (S), p-coumarol (H), and coniferyl alcohol (G) [16]. In previous studies, gymnosperms mainly contained H-type and G-type lignins, while angiosperms mainly contained S-type lignin and G-type lignin [17]. In the lignin biosynthesis pathway, the caffeic acid methyltransferase (*COMT*) gene is one of the key genes regulating lignin synthesis. It is primarily involved in the synthesis of S-type lignin [18]. It can be used to catalyze the methylation of caffeic acid, 5-hydroxyconiferaldehyde, and 5-hydroxyconiferyl alcohol to generate ferulic acid, sinusaldehyde, and sinapyl alcohol, respectively [19]. Analysis of gene structure and conserved domains showed that all *COMTs* have a c-terminal catalytic domain named Methyltrans-2, including a SAM/SAH binding pocket and a substrate binding site. Some of them show a common structure, the N-terminal domain, called dimerization [20]. The dimerization domain of LBD has been proposed, forming a helical structure, presenting a hydrophobic surface, which is formed by nine repeating heptapeptide motifs, containing hydrophobic residues at positions one and eight, and hydrophobic or charged amino acids with hydrophobic side chains at position five. Receptor monomer dimerization is a prerequisite for transcriptional activation and effective DNA binding of most nuclear receptors so far [21].

Previous studies have shown that dispersed duplication (DSD) and whole-genome duplication (WGD) are the main driving forces for the evolution of blueberry *COMTs*. In addition, the *COMT* of kiwi fruit has undergone two rounds of WGD [22]. In 1991, the *COMT* genes were cloned and described for the first time from *Ligusticum chuanqiong* rhizomes [23]. In the same year, the *COMT* gene was first identified as a gene family in *Populus tremuloides* [24]. It was later discovered that the *COMT* family is composed of multiple members. For example, the *COMT* gene family of *Eucalyptus grandis* has 7 *EgrCOMTs* genes [25]; the *COMT* gene family contains 23 members in *Catalpa bungei* [26]; there are 92 members in blueberries [20]; there are 25 *COMTs* genes in *Populus trichocarpa* [27]; there are 25 *COMTs* genes in *Brassic rapa* L. [28], and there are 25 *COMTs* genes in *Betula pendula* [29]. In this way, the function of the *COMT* gene family was gradually being elucidated. For instance, the *COMT* gene is highest expressed in the stems of the spray cut chrysanthemum, which is related to the rigidity of the pedicel [30]. The lignin concentration of *Brassica napus* was considerably decreased after *COMT* gene expression was inhibited, but the seed oil content and quality were unaffected [31]. It can be seen that the *COMT* gene plays an important regulatory role in the process of plant lignin synthesis. Furthermore, the *COMT* genes were shown to be involved in the biosynthesis of melatonin, which might improve the salt tolerance of *Arabidopsis thaliana* [32]. In short, *COMT* family members play an important regulatory role in plant lignin synthesis and have great potential in lodging and stress resistance.

The *COMT* gene’s well-established function provides a strong framework for our study, but a more comprehensive genome-wide investigation of the *COMT* gene family in soybean has yet to be completed. In this study, the *COMT* gene family in the soybean genome was analyzed at the genome-wide level, and a total of 55 *COMT* genes were identified. Then, we investigated the phylogenetic relationship, gene structure, protein motifs and domains, and chromosomal location of these *GmCOMTs* genes, and the cis-acting elements in the promoter region of the *COMT* genes. Moreover, the tissue specificity of *GmCOMTs* gene expression and its expression under different stresses were also explored. The research results will help explore the biological functions of the *COMT* gene family in soybean and further improve soybean yield and quality.

## 2. Results

### 2.1. Identification of COMT Gene Family in Soybean

To find all possible members of the *COMT* gene family in soybean, the known *Arabidopsis COMT* protein sequences were used as query sequences in a BlastP search against the soybean genome database. Finally, 55 *COMT* genes were identified, which were named as *GmCOMT1~GmCOMT55*, respectively. The *COMT* genes length, amino acid sequence of genes, genomic sequence, transcript sequence, and genomic location were also determined (Appendix A). The coding sequence length of the 55 *GmCOMTs* identified varied from 330 bp (*GmCOMT30*) to 1149 bp (*GmCOMT24*), the genomic sequence length ranges from 933 bp (*GmCOMT19*) to 9540 bp (*GmCOMT7*), and the transcript sequence length ranges from 330 bp (*GmCOMT30*) to 3295 bp (*GmCOMT34*). The length of encoded protein varies between 109 and 382 amino acids. Furthermore, the molecular weight of 55 *GmCOMTs* was between 12,721.95 and 42,846.58, and the highest molecular weight protein was *GmCOMT24*; the smallest molecular weight is *GmCOMT30*. The predicted isoelectric points range from 4.85 (*GmCOMT21*) to 9.07 (*GmCOMT28*) (Appendix A).

### 2.2. Phylogenetic Analysis of the COMT Gene Family in Soybean

The phylogenetic tree was constructed using the full-length protein sequences of 55 *Glycine max COMTs*, 14 *Arabidopsis thaliana COMTs*, 17 *Solanum lycopersicum COMTs**,* 48 *Oryza sativa COMTs,* and 25 *Brassic rapa COMTs* (Figure 1). The numbers of *COMTs* genes in *S. lycopersicum*, *A. thaliana,* and *B. rapa* in different groups are shown in Appendix A. Our study showed that the *COMTs* in these four species could be classified into group I and group II. Group I had 35 soybean *COMTs* and 12 tomato *COMTs*, whereas group II contained 20 soybean *COMTs* and 5 tomato *COMTs*. The *COMT* proteins of soybean and *S. lycopersicum* were distributed in both groups, but the *COMT* proteins of *A. thaliana* and *B. rapa* were only distributed in group II. In summary, the phylogenetic tree illustrates *COMTs’* evolutionary constancy across species. The analysis of the biological activities of these proteins is made possible by determining the homology connection between species.

### 2.3. Evolutionary Conservation Analysis of COMTs in Soybean

The phylogenetic tree was constructed with 55 *COMT* protein sequences of soybean (Figure 2a). Phylogenetic analysis of *GmCOMTs* shows that 55 *GmCOMTs* gene sequences can be divided into two groups: group I contains 35 *GmCOMT* proteins, and the remaining 20 *GmCOMT* proteins are in group II.

The evolution of numerous gene families is aided by the diversity of gene structure [33]. The obtained gene structure reveals significant variations in the variety of GmCOMTs gene structures (Figure 2b). In *GmCOMTs,* family of the same group have similar sizes and introns quantities. The majority of the *GmCOMTs* have 4–7 introns; however, *GmCOMT5*, *GmCOMT19*, *GmCOMT30,* and *GmCOMT32* each have just 3 introns, and *GmCOMT49* has 8 introns. Furthermore, *GmCOMT7* is the longest gene, while *GmCOMT19* is the shortest gene, and there are 16 *GmCOMTs* genes without a complete UTR region.

Next, we utilized the MEME website (http://meme-suite.org/; accessed on 9 July 2021) to predict protein motif and further investigate the structural diversity of *GmCOMT* proteins [34] (Figure 2c). In the analysis of conservative motifs, a total of 10 conserved motifs in the *GmCOMTs* proteins were identified, each of which was shown distinct, respectively (motif 1~motif 10). The width, sites, and E-value of conservative motifs in *GmCOMTs* protein were shown in Appendix A. In the evolutionary tree, the two sets of *GmCOMTs* genes share comparable conserved patterns. However, motifs 3 (IIHNHGQPITLSELVSSLQIPPSKACFVQRLMRFLAHNGFF) were identified solely in group I, as opposed to group II. Most members of the soybean *COMT* family contain comparable conserved motifs, indicating that their motifs are similar in composition and may have similar biological functions.

Further analysis of the conserved domains of soybean *COMTs* protein revealed that all 55 *GmCOMTs* genes contained Methyltransf-2 domains (Figure 2e). In addition, the dimerization domains are distributed in 48 *GmCOMTs* genes; dimerization 2 domains only exist in *GmCOMT46*.

Although these 55 *GmCOMT* genes have little difference in domains, so as to ensure the conservation of their functions, there are obvious differences in gene structure and motifs, and they are divided into two groups, namely group I and group II (Figure 2a).

### 2.4. Chromosomal Location and Collinearity Analysis of COMT Gene FAMILY in Soybean

The location of *GmCOMT* among 20 chromosomes was displayed using MapChart software based on the position information of 55 *COMT* genes in the soybean genome (Figure 3), and the specific position information can be checked in Appendix A. The 55 *GmCOMTs* genes are found on chromosomes 2, 4, 6–15, and 18–20, respectively, with no *GmCOMT* on the other five chromosomes (1, 3, 5, 16, 17). The number of *GmCOMTs* genes found on each chromosome ranged from one to seven. Only one gene was found on chromosomes 2 and 19, whereas seven were found on chromosomes 6 and 20. The findings reveal that *GmCOMTs* genes are not uniformly distributed on soybean chromosomes, with roughly 67% clustered at the bottom of the chromosomes. It is worth mentioning that several *GmCOMTs* genes that are grouped on the same chromosome are also found in the same phylogenetic tree group. For example, the four genes on chromosome 20 (*GmCOMT1*, *GmCOMT2*, *GmCOMT4,* and *GmCOMT34*) belong to group I of the phylogenetic tree, while the three genes *GmCOMT53*, *GmCOMT54*, and *GmCOMT55* belong to group II. This occurrence suggests that the *GmCOMTs* genes’ aggregated distribution may have comparable molecular functions, and the functional annotations of the *GmCOMTs* genes in soybean are available in Appendix A. The studies on soybean and wild soybean genomes showed that 13 million years ago, the whole genome duplication (WGD) event expanded the soybean genome [35]. During the long-term domestication of soybeans, the redundant genes generated by this WGD event declined. This phenomenon was confirmed by the soybean WGD (Appendix A), in which the collinearity genes in the collinearity region occupies most of the repetitive genes. (Appendix A). However, most of the *COMT* family genes we predicted also belong to this type of doubling (Appendix A), which indicated that the redundancy of *COMT* family genes was accompanied by a genome-wide WGD event.

Eventually, the collinearity of *GmCOMTs* genes is explored by using MCScanx and TBtools software (Figure 4a). To explore the potential evolution relationship and further compare the *COMT* gene family collinearity among different species, a comparative analysis between the *GmCOMT* protein and homologues from the other five representative plants, including *Populus trichocarpa*, *Betula Pendula*, *Solanum lycopersicum*, *Arabidopsis thaliana,* and *Oryza sativa*, was performed (Figure 4b). The results showed that there are six (10.9%), seven (12.7%), five (9.1%), two (3.6%), and no (0.0%) *GmCOMT* proteins showing high homology to the members from the other five species, respectively (Figure 4b). Conclusively, the *COMT* gene family in soybean had more collinearity with woody plants (*Populus trichocarpa* and *Betula Pendula*) but showed a little intersection with herbaceous plants (*Solanum lycopersicum*, *Arabidopsis thaliana,* and *Oryza sativa*), indicating that the *COMT* gene family had undergone evolutionary divergence in woody and herbaceous plants. The Ka/Ks ratios of the homologous genes measured in the *COMT* gene subfamily were all less than 1 (Figure 4c), indicating that the *COMT* gene family undergoes purifying selection during evolution, which helps to understand the evolution of the soybean *COMT* gene family. Furthermore, the divergence time of most soybean *COMT* genes is within 10 MYA (Appendix A), which is later than that of *Arabidopsis* (9.6–16.1 MYA) [36].

### 2.5. The Cis-Regulatory Elements in the Promoter of GmCOMTs Genes in Soybean

The PlantCare website (http://bioinformatics.psb.ugent.be/webtools/plantcare/html/; accessed on 13 April 2021) was used to predict and analyze the cis-regulatory elements of the 2000 bp nucleic acid sequences upstream of the *GmCOMTs* genes transcriptional start site in order to investigate the cis-acting elements found in the soybean *COMT* gene family and their possible regulation (Figure 5). The cis-acting elements in the promoter region of *GmCOMT* genes can respond to various biotic and abiotic stresses, therefore regulating the expression of downstream genes. The results showed that the cis-acting elements on the promoter region of the *GmCOMT* gene family are mostly split into four classes: growth and development, light response, stress response, and hormone response, as shown in Appendix A.

With 2194 CAAT-boxes found in virtually all *GmCOMTs* genes, it is the most predicted of all cis-elements. The light response is represented by 20 cis-acting elements, accounting for roughly 36% of all cis-acting elements, suggesting that soybean *COMT* genes are likely to be controlled by light. We also found ABRE, TCA, as-1, ERE, TATC-box, TGA-box GARE-motif, TGACG-motif, and AuxRR-core, as well as nine cis-acting elements linked to hormones. Furthermore, 16 different cis-acting elements linked to stress response were identified. For example, 21 MBSs associated with low-temperature response, 102 AREs related to drought stress, 22 TC-rich repeats related to salt stress, 16 LTRs related to anaerobic stress, etc. These comprehensive findings suggest that *GmCOMTs* can participate in a variety of stress responses. These comprehensive findings suggest that a range of stressors or external stimuli can influence the expression of the *GmCOMT* genes.

### 2.6. Tissue Specificity of GmCOMTs Genes Expression

To address the expression patterns of the *COMT* gene family in soybean, two representative soybean varieties, Jack and Williams82, were selected for different tissues at the VC stage (after 14 days of seedlings, two unifoliate leaves unroll), including epicotyls, hypocotyls, meristems, roots, and unifoliate leaves. The findings revealed that most *GmCOMT* genes were strongly expressed in the roots, with 29 of them being the same (Figure 6a,b). In the Jack variety, there were 5, 7, 11, and 5 highly expressed genes in the epicotyl, hypocotyl, meristem, and unifoliate leaves, respectively. In Williams82, there were 3, 8, 10, and 8 highly expressed genes in epicotyl, hypocotyl, meristem, and unifoliate leaves. The expression results showed that *COMTs* were mainly expressed in roots, and the expression levels were similar in Jack and Williams82. At the same time, we can see that the two genes, *GmCOMT6* and *GmCOMT7*, are strongly expressed in the hypocotyl of the Jack variety; however, the expression level is low in the hypocotyl of the Williams82 variety but highly expressed in the unifoliate leaves. The study’s findings revealed that the expression of the same gene differs amongst varieties. Furthermore, for verification the data of RNA-seq, three genes (*GmCOMT25*, *GmCOMT33,* and *GmCOMT53*) for RT-qPCR were performed to evaluate the expression level of three genes in the epicotyls, hypocotyls, meristems, roots, and unifoliate leaves of Jack and Williams82, respectively (Figure 6c); it can be seen that *GmCOMT25*, *GmCOMT33,* and *GmCOMT53* genes are highly expressed in the leaves, roots, and meristems of Jack and William, respectively. The results show that it is consistent with the transcriptome.

### 2.7. Expression Patterns of COMT Genes under Abiotic Stresses

The expression patterns of *COMTs* were investigated under drought, submergence, dehydration, and salt stresses (Figure 7). After 5–6 days of drought treatment, the expression of 11 *GmCOMTs* genes was considerably decreased compared to the control. However, following 5 days of drought treatment in the roots, *GmCOMT19* expression increased significantly. The expression of most *GmCOMTs* genes was considerably decreased after submergence treatment. Gene expression was generally reduced during drought and floods, but it was restored after one day of recovery. Compared to the control, the expressions of seven genes were considerably enhanced after 12 h of dehydration treatment. The expressions of *GmCOMT29* and *GmCOMT34*, on the other hand, were considerably decreased. The expression of 18 genes rose substantially after 6 or 12 h of salt treatment. The findings revealed that the majority of *GmCOMT* genes differed depending on the stress treatment.

## 3. Discussion

There was abundant evidence that soybeans were an important food crop on Earth, providing us with a rich source of protein. For plants, lignin was an important polymer compound with many biological functions [37]. If the content of lignin in the plant was reduced, the plants would be in a lodging state [38], reducing the ability of water transport and mechanical support, and it was not conducive to resisting various external pressures, to pose a serious threat to the growth and development of the plants [39]. Reduced lignin concentration in soybeans weakens the strength of the stalks and reduces resilience to pests, diseases, and pathogens, according to previous research [40,41]. The biosynthesis of lignin was an extremely complex and delicate process formed by the polymerization of aromatic structural unit phenylpropanine [42]. It was generally believed that lignin comes from three important monomers, p-hydroxyphenyl (H), syringyl (S), and guaiacyl (G) units [43]. Among them, the *COMT* gene was a key gene that regulates the synthesis of S-type lignin. So far, the function of the *COMT* gene family in soybeans was still unclear. Therefore, we preliminarily predicted the function of *GmCOMTs* based on the homology of other species.

In *Arabidopsis thaliana*, the *At5G54161.1* (O-methyltransferase 1) gene was shown to be highly essential in the production of S-type lignin [44] and catalyzed the methylation of n-acetylserotonin to produce melatonin [45]. Melatonin was considered a growth-promoting compound and rooting agent [46] which played an important role in the growth and development of plants. In addition, melatonin made a significant contribution to improving the stress tolerance of plants [47]. Our results showed that *AT5G54160.1* was closely related to the five genes *GmCOMT51*, *GmCOMT52*, *GmCOMT53*, *GmCOMT54*, and *GmCOMT55* in group II, indicating that *GmCOMTs* in group II may have similar functions. In general, *COMT* is considered to be mainly involved in the lignin synthesis pathway, converting 5-hydroxy-coniferyl aldehyde to sinapisaldehyde. However, recent studies have shown that *COMT* can participate in the final step of catalyzing the synthesis of melatonin in watermelon, and we speculate that the functions of different *COMT* gene family members are differentiated.

We can observe from the gene structure study results that the majority of *GmCOMTs* genes have numerous introns. In the phylogenetic tree, the *GmCOMTs* in the same group showed very comparable intron–exon architectures. The number of introns in the *GmCOMTs* gene in group II was typically higher than in group I, suggesting that the *COMT* genes with the large number of introns may have a better selective advantage in evolution. The Methyltransf-2 domain was found in all 55 *GmCOMTs* gene sequences, with the majority of the genes having just one Methyltransf-2 domain; however, two genes (*GmCOMT9* and *GmCOMT19*) had two Methyltransf-2 domains. In previous reports, the *PtrCOMTs* gene sequence of *Populus trichocarpa* contains only one Methyltransf-2 domain [48]; some blueberry *VcCOMTs* genes contain two or three Methyltransf-2 domains [20]. The 55 *GmCOMT* protein sequences all contain Methyltransf-2 domain, so they are relatively conservative; meanwhile, in the analysis of motifs, most of the motifs are relatively conservative in the two groups. For example, some residues of Motif 2 (‘VGGGTGTTAKIICEAFPKJKCIVFDLPHV’) are related to the SAM/SAH binding site [49], which is consistent with the previous *COMTs* verification of blueberry and alfalfa. However, some motifs are specific. For example, Motif 3 (‘IIHNHGQPITLSELVSSLQIPPSKACFVQRLMRFLAHNGFF’) only exists in the group I of *GmCOMTs* protein sequence and is not included in group II; therefore, it is consistent with the classification analysis of the evolutionary tree. Based on the analysis of conservative motifs, it is found that the presence or absence of Motif 3 in the soybean *COMT* protein sequences is one of the important reasons for the classification based on the phylogenetic tree. However, there is no difference in the conserved domains of the two groups; therefore it is consistent with the classification analysis of the evolutionary tree. Thus, this conserved sequence might play an important role in maintaining the structure and function of *COMT* in soybean.

According to cis-acting element findings, the majority of cis-acting elements linked to light response were discovered in soybean *COMT* genes. There were 128 G-boxes, 63 GT1-moths, 23 GATA-moths, and so on, all of which were important in light-mediated transcriptional activity. Therefore, the *GmCOMTs* gene was likely to be influenced by the light-induced proteins. Meanwhile, we also discovered many cis-acting elements related to stress. In previous studies, the *COMT* genes were induced by salt stress and affected by cold stress [50]. These stress-related elements, such as ARE (related to drought), TC-rich (related to salt stress), and MBS (related to low temperature), all indicated that the *GmCOMTs* were likely to be induced by abiotic stress. According to reports, transgenic *Arabidopsis* overexpressing *CrCOMT* genes showed better physiological performance than wild type under salt stress [51]; and the expression pattern of *COMT* gene family under salt stress can infer that *GmCOMTs* genes in group II have similar functions.

Finally, the expression patterns of the *COMT* gene family in different soybean tissues are similar. Among them, the *COMT* gene family has the highest expression in the roots of Jack and Williams82. However, the differences in the expression patterns of different tissues in the same breed indicate that there are functional differences between the *COMT* gene families. The *COMT* gene family may regulate gene expression in response to dehydration and salt stress in soybean. Therefore, the *COMTs* expression of W82 roots under these stress conditions was analyzed. Our results indicated that, under salt stress condition, the expression of *COMTs* increased. However, under drought and submergence stress conditions, the expression of *COMTs* in soybean was significantly reduced. At the same time, the expression of *COMTs* in different tissues from distinct soybean varieties showed that the expression of *COMTs* in the roots was higher. We also used the W82 leaves and roots of the V1 stage to verify. It is interesting to find that the gene expression levels in the roots on the first day of stress treatment increased significantly. Regarding the mechanism of this phenomenon, it also needs further study. Therefore, in this study, the family structure, evolutionary relationship, and expression pattern of *GmCOMTs* genes were comprehensively analyzed, which played a vital role in soybean stress resistance and provided a theoretical basis for its future breeding.

## 4. Materials and Methods

### 4.1. Plant Materials

Two soybean varieties, Jack and Williams82, were grown as plant materials in the laboratory greenhouse of Northeast Forestry University (26 °C, 14 h light/10 h dark). When the seedlings were 14 days old (VC stage, two unifoliate leaves unroll), different tissues of the plant, including the epicotyls, hypocotyls, meristems, roots, and unifoliate leaves, were collected, with three independent copies of each sample. The collected plant material was frozen in liquid nitrogen and stored at −80 °C.

### 4.2. Identification of COMT Gene Family in Soybean

The candidate protein sequences were identified using the known model plant-*Arabidopsis COMT* gene and soybean genomic protein database for BlastP analysis (E-value threshold <= 1 × 10^−5^). Furthermore, 55 soybean *COMT* genes were identified, using HMMER 3.1 (http://hmmer.org/download.html; accessed on 23 March 2021) software to screen candidate genes through hidden Markov models [52]. The isoelectric points and molecular weights of all soybean *COMT* gene families were obtained online on the ExPASy (http://www.expasy.org/tools/; accessed on 18 September 2021) website [53], and the parameters were set to average values.

### 4.3. Phylogenetic Analysis of COMT Proteins in Soybean

The protein sequences of *COMT* from *G. max*, *A. thaliana*, *B. rapa, O. Sativa* and *S. lycopersium* were subjected to multiple sequence alignment, and the phylogenetic tree was built using the Neighbor-Joining (NJ) method of MEGA-X software 10.0 [54,55]. The robustness of each node in the tree was determined using 1000 bootstrap replicates, and the default parameter for the remaining parameters was selected. The visualization of the evolutionary tree is achieved through the iTOL website (https://itol.embl.de/; accessed on 20 November 2021).

### 4.4. Gene Structure, Motif and Domain Analyses for the GmCOMT Genes

The gene structures of the *COMT* gene family members in soybean were visualized using the soybean genome’s annotation information using TBtools software (version 1.092) [56]. The conserved motifs of soybean *COMT* proteins were analyzed by the MEME website (http://meme-suite.org; accessed on 9 July 2021) [34]. The MEME (http://meme-suite.org; accessed on 9 July 2021) website was used to analyze the conserved motifs of GmCOMT proteins with the following parameters, setting 10 motifs.

### 4.5. Chromosomal Location, Collinearity Analysis, Gene Duplication Events, and Ka/Ks Analysis of GmCOMT Genes

The chromosome mapping information of the *COMT* gene family in soybean was gained from the NCBI database, and the map was drawn by MapChart software [57]. The collinear relationship of *COMT* gene families between soybean and poplar was analyzed by using MCScanx and TBtools software (version 1.092) [58]. The gene duplication of *COMT* genes was characterized using BlastP and Multiple Collinearity Scan toolkit (MCScanX) (http://chibba.pgml.uga.edu/mcscan2/#tm; accessed on 10 July 2021) [59], and the synteny analysis of *COMT* genes among the *Populus trichocarpa*, *Betula pendula*, *Solanum lycopersicum*, *Arabidopsis thaliana,* and *Oryza sativa* was performed in TBtools with default parameters.

The Ka value, Ks value, and Ka/Ks ratios for the paralogs *COMT* gene pairs were calculated by TBtools. The paralogous genes were identified by searching the term “syntenic region” in the soybean genome. The rate of divergence was calculated by using the following formula: T = Ks/2r, where Ks represents the synonymous substitutions per site and r is the rate of divergence. For dicotyledonous plants, the hypothesis is 1.5 synonymous substitutions per site of 108 years [60].

### 4.6. Prediction of Cis-Acting Elements of COMT Genes in Soybean

The upstream 2000 bp genomic DNA sequences in the promoter regions of *COMT* genes were retrieved from the soybean genome using the TBtools software (version 1.092). The cis-regulatory elements (CREs) in the 2000 bp upstream regions of the *COMT* gene promoter were analyzed using the PlantCare website (http://bioinformatics.psb.ugent.be/webtools/plantcare/html/; accessed on 13 April 2021) [61,62].

### 4.7. Expression Pattern Analysis

The transcriptome data of soybean under diverse stress treatments (drought, submergence, dehydration, and salt) was obtained from the NCBI database (https://www.ncbi.nlm.nih.gov; accessed on 13 July 2021) with accession number PRJNA246058 (https://www.ncbi.nlm.nih.gov/bioproject/PRJNA246058/; accessed on 13 July 2021) and PRJNA574626 (https://www.ncbi.nlm.nih.gov/bioproject/PRJNA574626/; accessed on 13 July 2021). The obtained transcriptome data were processed and drawn into heat maps. The fragments per kilobase per million (FPKM) value was calculated to quantify gene expression.

The bioinformatics analysis process is as follows: (a) the original transcriptome data was aligned to the soybean genome with hisat2 (https://daehwankimlab.github.io/hisat2/; accessed on 20 July 2021). (b) The aligned results were performed to calculate transcripts per million (TPM) by stringtie (http://ccb.jhu.edu/software/stringtie; accessed on 20 July 2021). (c) Finally, the *GmCOMT* gene expression in different tissues was scaled by setting the scale = “row” parameter in pheatmap function of R (https://cran.r-project.org; accessed on 20 July 2021), and then the hierarchical clustering of expression profiles of 55 COMT genes was visualized by R-heatmap package.

### 4.8. Genomic RNA Extraction and Quantitative RT-PCR Analysis

TRIzol (TianGen, Beijing, China) was used to extract total RNA from each tissue of soybeans. The SYBR Green I Master mixture (Roche, Basel, Switzerland) was used as the reagent for qRT-PCR. The qRT-PCR primers designed in this study are shown in Appendix A. The 2^−ΔΔCT^ method was used to calculate the relative gene expression levels [63]. About 1 μg RNA was mixed with the 10 μL reverse-transcribed buffer in 2 mL PCR-tube, which comprised 1 μL gDNA Eraser Buffer, 2 μL 5 × *g* DNA Eraser Buffer, and the rest is RNase free ddH_2_O. The standard protocol was set as 2 min at 42 °C, followed by 10 min at 0 °C. The reaction product was added to a 10 μL reaction containing 1 μL Primer Script RT Enzyme, 1 μL RT Primer Mix, 4 μL 5 × Primer Script Buffer, and RNase Free ddH_2_O. The reverse-transcribed parameters were as follows: 15 min at 37 °C followed by 5 s at 85 °C and 10 min at 0 °C. The ten-fold diluted digestion DNA products were used as a template for quantitative RT-PCR of betulin synthetic genes. The tubulin gene was identified from the birch reference genome as the internal control. PCR analysis was conducted using the Applied ABI7500 Real-Time PCR System with SYBR Premix Ex Taq™ II. Each PCR was conducted in a 15 μL reaction mixture containing 6.3 μL of 20 × diluted cDNA, 7.5 μL of SYBR Green Supermix, 0.6 μL of forward primers, and 0.6 μL of reverse primers. Each gene of transcript level was calculated by the 2-(^−ΔΔCT^) CT method. All the qRT-PCR experiments were performed in three independent replicates.

## 5. Conclusions

A total of 55 *COMTs* genes were identified in the entire soybean genome, and these genes were distributed in two groups of the phylogenetic tree. *COMTs* in soybeans has large structural differences, but it is relatively conservative in evolution. In addition, the cis-acting elements in the promoter region of the *GmCOMTs* gene were predicted, showing a total of four types of elements. The *GmCOMTs* gene expression patterns in different tissues and stress conditions were systematically studied. It was found that the expression of soybean *COMT* gene family members had tissue differences and was related to abiotic stresses such as drought and salt. The systematic exploration of the soybean *COMT* gene lays the foundation for future soybean breeding and stress resistance.

## Figures and Tables

**Figure 1 plants-10-02816-f001:**
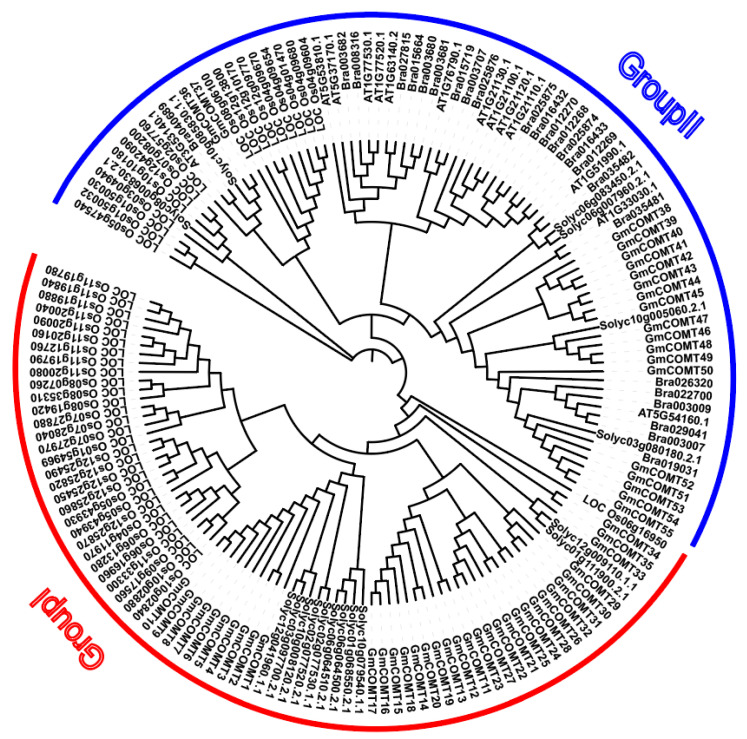
Phylogenetic relationship of *COMTs* in different species using complete protein sequences. The neighbor-joining (NJ) phylogenetic tree was constructed with Poisson model by MEGA-X 10.0 software. Gm represents *G. max;* Bra represents *B. rapa;* Solyc represents *S. lycopersium;* AT represents *A. thaliana,* Os represents *Oryza sativa,* respectively. The tree was generated from an amino acid sequence alignment of 55 *GmCOMTs*, 25 *BrCOMTs*, 17 *SlCOMTs**, 48 OsCOMTs,* and 14 *AtCOMTs*. The red arc represents group I; the blue arc represents group II, respectively. The tree is based on homologous groups showing evolutionary relationships with *COMTs*.

**Figure 2 plants-10-02816-f002:**
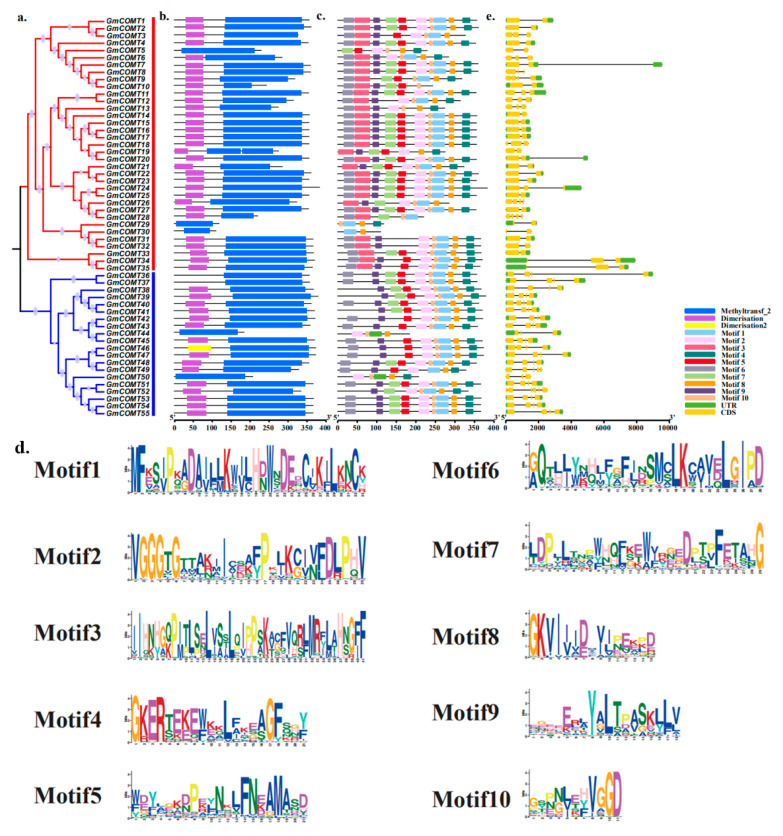
The polygenetic relationship, gene structure, and motif analysis of the *COMTs* in soybean. (**a**) The phylogenetic tree was constructed with the MEGA-X 10.0 program using protein sequences of the 55 *COMTs* genes in soybean. (**b**) Identification of the conserved functional domains of the *COMT* gene family of soybean. Blue represents the Methyltransf-2 domain, yellow represents the dimerization domain, and pink represents the dimerization 2 domain. (**c**) The size of the 55 *COMTs* were characterized by the MEME website and TBtools software, respectively. A total of 10 putative motifs are identified here, with the numbers 1–10, and are represented by different color boxes, which are numbered and placed at the bottom on the right. (**d**) Motif LOGO showing below. (**e**) Exon/intron structure analysis of 55 putative *COMT* genes. The yellow box is indicative of the exons (CDS), the green box represents UTR, and the grey line is described as the introns. The sizes of the exons or introns can be calculated from the bottom scale.

**Figure 3 plants-10-02816-f003:**
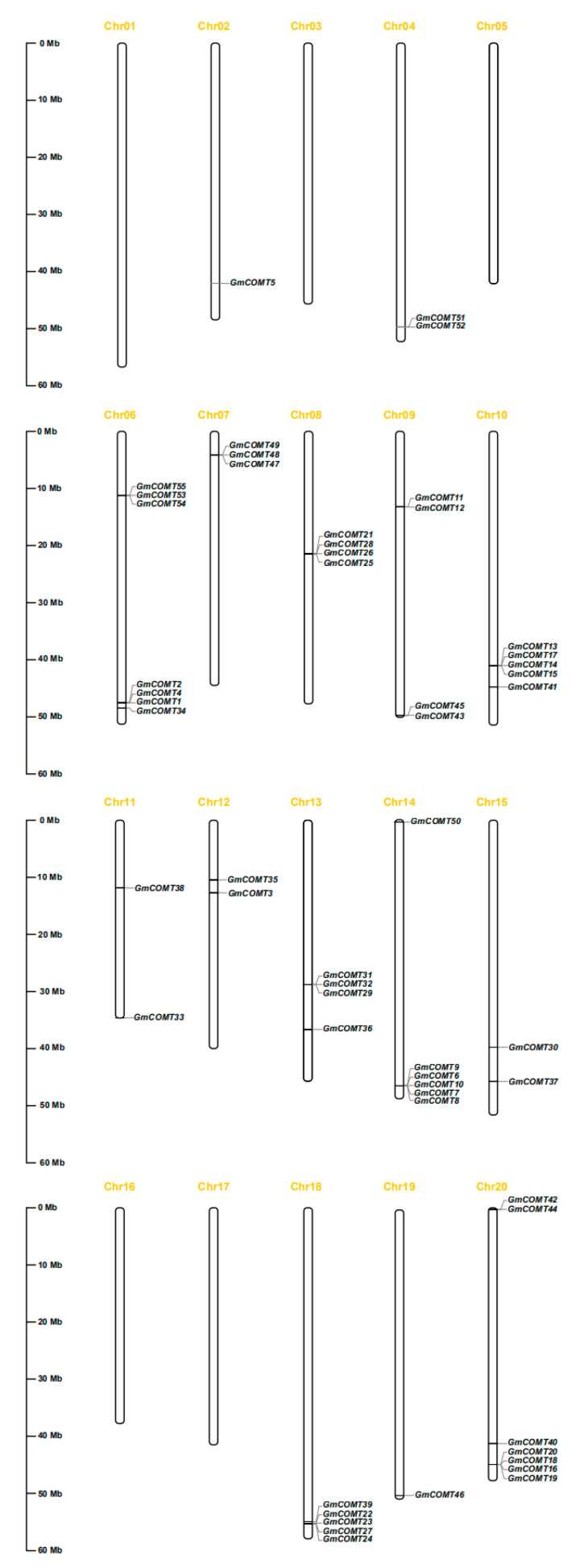
Chromosomal localization of *COMT* genes in soybean. There were 55 *COMT* genes located on 15 soybean chromosomes. Chromosome numbers are labeled at the top of each vertical bar. The left scale rulers indicate the length of the chromosomes. The names of the *COMT* genes are labeled at the appropriate position on the right side of each soybean chromosome.

**Figure 4 plants-10-02816-f004:**
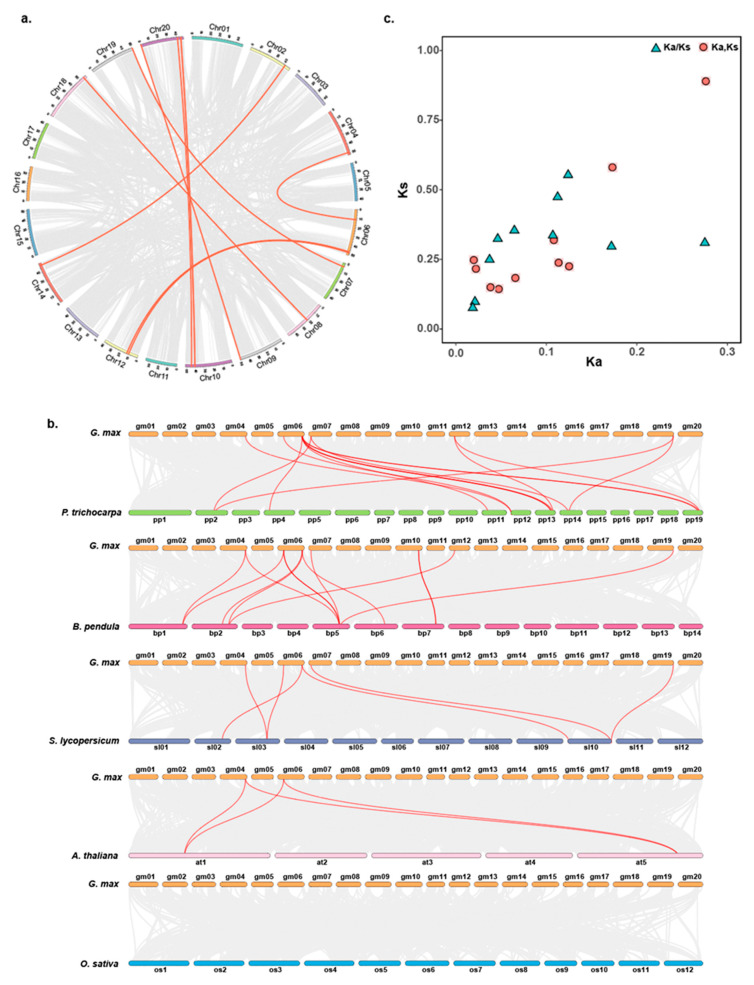
(**a**) Collinearity analysis of *COMT* gene family in soybean. Each colored square on the edge of the circle represents soybean chromosomes. The gray lines in the middle represent the soybean genome’s collinearity module, and the red lines show that some of the *COMTs* genes have a collinear relationship. (**b**) The homology analysis of *COMT* genes between soybean and five representative plant species. The gray lines represent the collinear blocks in the genome of soybean and other plants, and the red line represents the collinear *COMT* gene pairs. The plant named different prefixes “*G. max*”, “*P. trichocarpa*”, “*B. pendula*”, “*S. lycopersicum*”, “*A. thaliana*”, and “*O. sativa*” represent *Glycine max*, *Populus trichocarpa*, *Betula pendula*, *Solanum lycopersicum*, *Arabidopsis thaliana,* and *Oryza sativa*. (**c**) The ratio between Ks and Ka for paralogous *GmCOMT* gene pairs in soybean.

**Figure 5 plants-10-02816-f005:**
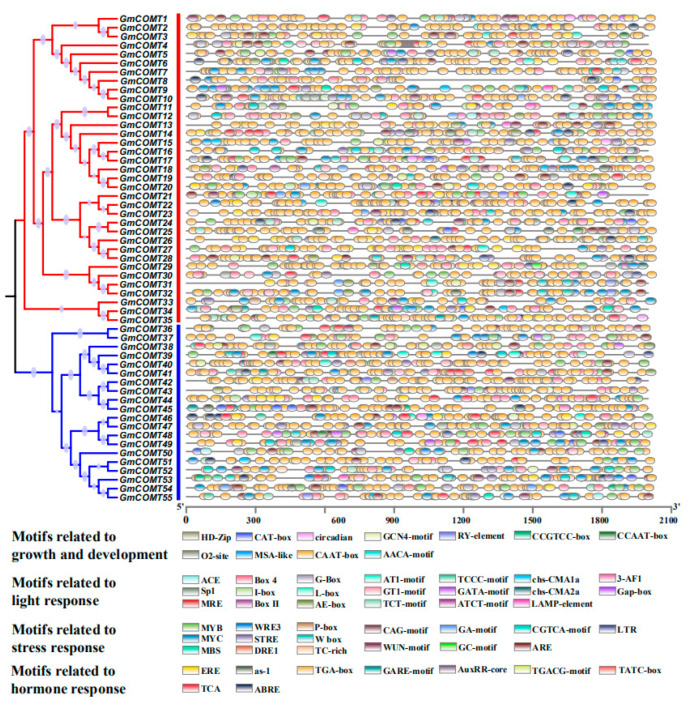
Predicted cis-regulatory elements in the 2000 bp promoter regions of *COMT* genes in soybean. Putative core sequences are represented by different symbols, as indicated in the figure key at the bottom. Different color squares represent different cis-acting elements.

**Figure 6 plants-10-02816-f006:**
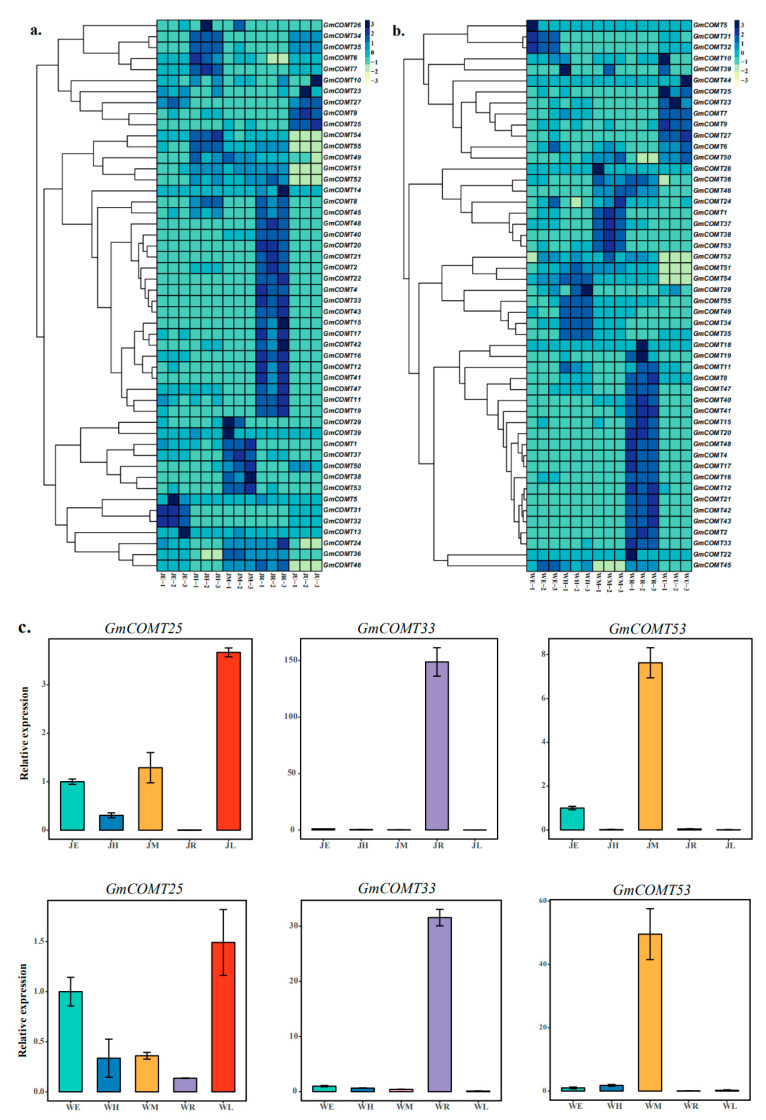
The expression levels of *GmCOMTs* genes in Jack (**a**) and Williams82 (**b**). The colors spanning from green to blue represent an increasing level of gene expression. The horizontal axis represents the different tissue parts of the soybean, and the vertical axis represents the *COMTs* genes of soybean. The expression patterns of *GmCOMT25*, *GmCOMT33,* and *GmCOMT53* are shown in Jack and Williams82 (**c**). Different colors represent different tissues. E, epicotyl; H, hypocotyl; M, meristem; R, roots; U, unifoliate leaves.

**Figure 7 plants-10-02816-f007:**
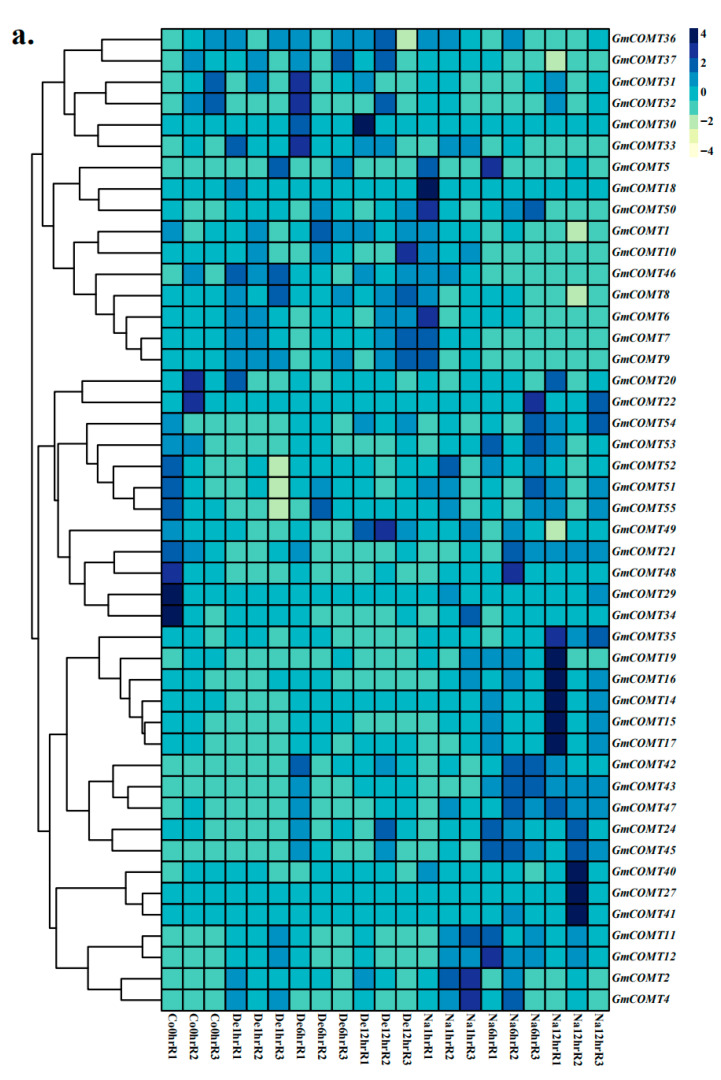
Expression patterns of *GmCOMTs* genes under different abiotic stresses, (**a**) dehydration and salt stress, (**b**) drought and submergence. Co and CT represent control treatment; De and Na represent dehydration and salt stress, respectively; DRO and SUB represent drought and submergence. D and h represent day and hour. L and R are the leaves and roots, respectively. DRO_REC_L/R indicates one-day recovery following six days of drought in leaves/roots. SUB_REC_L/R indicates one-day recovery following three days of submergence in leaves/roots.

## Data Availability

All data on NCBI database (https://www.ncbi.nlm.nih.gov; accessed on 13 July 2021) is publicly available.

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
