# Peer review of "Genome-Wide Identification and Characterization of *Caffeic Acid O-Methyltransferase* Gene Family in Soybean"

_plants, 2021, doi:10.3390/plants10122816_

Round 1
Reviewer 1 Report
The manuscript by Zhang et al. deals with Caffeic acid O-methyltransferases (COMTs) family genes in soybean. The COMTs genes play an essential role in regulating lignin synthesis. Lignin is one of the major components of the cell wall and is directly associated with plant development and stress management. Although COMTs family genes have been characterized in many plants, it is not yet explored in soybean. A total of 55 COMT genes in soybean (GmCOMTs) were identified and further investigated in relation to stress, such as salt and drought stress.
The authors have made several modifications in this current version; however, the flow of writing needs to be checked, avoiding sentence repetition.
please include the respective p values in fig 2a and E values, site and width in 2d
This will be interesting to see the expression of GmCOMT25, GmCOMT33 and GmCOMT53 in developmental stages
Is there any protein-protein interaction among of GmCOMTs
Please check the format of the reference.
Author Response
Response to Reviewer 1 Comments
Point 1: The authors have made several modifications in this current version; however, the flow of writing needs to be checked, avoiding sentence repetition.
Response 1: Yes, we have carefully checked and revised our manuscript.
Point 2: please include the respective p values in fig 2a and E values, site and width in 2d.
Response 2: Yes, we have added p values of Figure 2a in Table 1, E values, site and width of Figure 2d in Table S3. Because putting them in the text will affect the appearance.
Point 3: This will be interesting to see the expression of GmCOMT25, GmCOMT33 and GmCOMT53 in developmental stages.
Response 3: Thank you so much for your comments. The differential expression of soybean COMT genes in different periods can indeed reflect the role of these genes in lignin synthesis at different periods. But we do quantitative experiments to verify the transcriptome, the main focus of this article is the identification of the GmCOMT gene family, gene structure and expression patterns. GmCOMT gene expression in different periods is not our focus, so we will verify and publish your opinions in subsequent experiments.
Point 4: Is there any protein-protein interaction among of GmCOMTs.
Response 4: The present study focuses on identifying the soybean COMT genes, analyzing the structure and expression patterns of the COMT genes, and the protein localization experiment is not the focus of our paper analysis, we will adopt opinions and further verify in the future research.
Point 5: Please check the format of the reference.
Response 5: Yes, we have done the formatting of the references in our manuscript.
Table1. The Newick fill of the neighbor-joining (NJ) phylogenetic tree.
((((((((((Glyma.10G176500.1:0.02716368,Glyma.20G213700.1:0.02248155)0.6990:0.00757380,Glyma.10G176700.1:0.03690292)0.9990:0.04388572,(Glyma.20G213600.1:0.10014010,(Glyma.20G213500.1:0.08847175,Glyma.20G213800.1:0.09238030)0.9880:0.04882900)0.2430:0.00163884)0.4740:0.01097680,Glyma.10G176600.1:0.09698094)0.7210:0.01905041,Glyma.10G176300.1:0.17311026)0.9050:0.02812061,(Glyma.09G094600.1:0.00228862,Glyma.09G094400.1:0.01725597)1.0000:0.13981862)1.0000:0.14031313,(((Glyma.18G267800.1:0.09804944,Glyma.18G267500.1:0.12867193)0.4730:0.01753271,Glyma.08G246700.1:0.13559909)0.3370:0.01864459,((Glyma.08G248000.1:0.05104794,Glyma.18G269600.1:0.07639701)1.0000:0.09391921,(Glyma.08G247000.1:0.06120907,(Glyma.08G246900.1:0.16772401,Glyma.18G267900.1:0.03851773)0.8890:0.03337930)0.9010:0.04845612)0.1500:0.00616300)1.0000:0.15625284)0.9500:0.05332165,(Glyma.13G173800.1:0.21331242,(Glyma.15G220700.1:0.28020088,(Glyma.13G173600.1:0.00000000,Glyma.13G173300.1:0.00000000)1.0000:0.11596183)0.7730:0.04998001)0.9990:0.16954157)0.9960:0.08006681,(((Glyma.12G119600.1:0.10537421,Glyma.06G286200.1:0.05496844)1.0000:0.14833379,Glyma.06G286700.1:0.26470529)0.6500:0.02903819,(Glyma.06G286600.1:0.15254429,(Glyma.02G233100.1:0.06286599,(Glyma.14G200800.1:0.00000000,(Glyma.14G200900.1:0.01974661,(Glyma.14G201100.1:0.00000000,(Glyma.14G201000.1:0.01968259,Glyma.14G200700.1:0.00000000)0.9680:0.03182517)0.5850:0.00793463)0.8950:0.02155616)0.7920:0.03697125)1.0000:0.09217922)0.9190:0.04478109)1.0000:0.16416921)0.9990:0.13857536,(Glyma.11G256500.1:0.49664961,(Glyma.12G109800.1:0.01490444,Glyma.06G295700.1:0.03068750)1.0000:0.43154160)0.5270:0.02417440,((Glyma.15G241100.1:0.03531331,Glyma.13G263200.1:0.02922521)1.0000:0.49224160,(((((Glyma.10G215700.1:0.05200539,Glyma.20G176100.1:0.05117884)1.0000:0.14935148,Glyma.18G263700.1:0.26649845)0.9630:0.06137211,Glyma.11G150800.1:0.29075208)1.0000:0.09737822,((Glyma.09G281900.1:0.01343371,Glyma.20G003500.1:0.03875204)1.0000:0.17338088,(Glyma.09G281800.1:0.16864830,Glyma.20G003600.1:0.05589287)0.9270:0.11026143)0.9990:0.09219804)0.9990:0.12871524,(((Glyma.07G048900.1:0.24906805,Glyma.19G260700.1:0.26633424)0.5110:0.02926409,(Glyma.07G048700.1:0.14115861,Glyma.07G048800.1:0.08278144)1.0000:0.17029293)0.9960:0.10295390,(Glyma.14G005000.1:0.12624012,((Glyma.04G227800.1:0.06841744,Glyma.04G227700.1:0.01136967)0.3750:0.00422047,(Glyma.06G137200.1:0.02251103,(Glyma.06G137100.1:0.01958347,Glyma.06G137300.1:0.01719578)0.6880:0.00747293)0.4060:0.00483322)1.0000:0.22530463)0.8920:0.07757577)0.8940:0.04505285)0.9770:0.08761411)0.9990:0.16029499);
Reviewer 2 Report
see comments to the editor and the comments posted in the first revision.
Thank you for resending the MS.
Does it seem that the authors failed to present WHAT NOVELTY THIS MS PRESENTS?
This study involves applied bioinformatics and adding a repository of genes in one species. Such a study should include a legitimate hypothesis with novelty not just because someone has not presented such a collection of genes in this species.
This type of information can nowadays be retrieved from relevant websites such as phytozome (JGI) and Soybase.
Author Response
Response to Reviewer 2 Comments
Point 1: Does it seem that the authors failed to present WHAT NOVELTY THIS MS PRESENTS?
This study involves applied bioinformatics and adding a repository of genes in one species. Such a study should include a legitimate hypothesis with novelty not just because someone has not presented such a collection of genes in this species.
Response 1: As an important national strategic economic crop resource, soybean yield and quality are directly related to the survival of all mankind [1]. To obtain good quality soybeans, the mechanism of lodging resistance and disease resistance is a very important research topic. Previous research has shown that wheat cultivars with greater lignin content were also shown to have improved lodging resistance [2]. However, little is known about the research on soybean-related lignin so far. Based on the existing lignin synthesis pathways, we know that the COMT gene family is a very important rate-limiting enzyme in the process of G/S lignin synthesis. Therefore, this study identified 55 soybean COMT family genes in soybeans for the first time through bioinformatics methods, and initially explored the characteristics of soybean COMT genes through transcriptome data from different tissues and multiple stresses. This research laid an important foundation for further research on the synthesis of lignin in soybeans.
[1] Pagano, M.C.; Miransari, M. The importance of soybean production worldwide. Abiotic and Biotic Stresses in Soybean Production. 2016, 1, 1-26.
[2] Tripathi, S.C.; Sayre, K.D.; Kaul, J.N.; Narang, R.S. Growth and morphology of spring wheat (Triticum aestivum L.) culms and their association with lodging: Effects of genotypes, N levels and ethephon. Field Crops Research. 2003, 84, 271-290.
This manuscript is a resubmission of an earlier submission. The following is a list of the peer review reports and author responses from that submission.
Round 1
Reviewer 1 Report
The manuscript by Zhang et al. deals with Caffeic acid O-methyltransferases (COMTs) family genes in soybean. The COMTs genes play an essential role in regulating lignin synthesis. Lignin is one of the major components of the cell wall and is directly associated with plant development and stress management. Although COMTs family genes have been characterized in many plants, it is not yet explored in soybean. A total of 55 COMT genes in soybean (GmCOMTs) were identified and further investigated in relation to stress, such as salt and drought stress.
The experimental methodology is correct and detailed presented; however, the interpretation and presentation of the results, including discussions, should be a bit more elaborated. Authors need to address some critical queries also, that are as follows
for ‘Identification of COMT gene family in soybean’ why only Arabidopsis COMT protein sequences as a query? Not Arabidopsis and rice.
What was the threshold of E-value for BlastP search??
Authors can also look for redundant sequences?? duplicated gene pairs?? tandem repeats for this genome annotation study
please explain more details about the methodology parameter and related criteria??
L105 between ‘12,721.95 and 42,846.58’ unit missing
Fig. 2 for motif analysis color representation is confusing and not explained appropriately
Fig. explanation of figure 4a and 4b is missing
Why for homology analysis of COMT genes soybean and poplar were chosen?
RT-qPCR experiment details are missing in the method section?
Please check line by line of references format.
Reviewer 2 Report
Dear authors,
Please consult the attached pdf for the comments. These comments are for the improvement of your story that will help you in the next submissions (here or elsewhere).
Overall, I do not see any novel information that enhances our understanding of the biological functioning of COMTs. The first known COMT was identified and characterized three decades ago. During the post-genomic era enormous information on the biological function of COMTs in plant development, tolerance to biotic and abiotic stresses has been made available. The authors should have explained clearly the SRA datasets in terms of functions.
This study lack a biological hypothesis. It is just a repository of COMTs in soybean.
Authors studied GmCOMTs in this report just because it hasn't been done already? See below some suggestions that can indicate the need of the study.
- Include multiple species from Kingdom Plantae that have differences in the structure of cell walls, from here we can expect that the COMTs might be evolved differently.
- Identify COMTs in the simple, and modern plants and compare them based on structure.
- Explore, what might have happened during evolution of plants class in terms of COMTs (and generally related to cell wall).
- Then you can develop an updated phylogeny of that should represent at least particular class of Kingdom plantae that have evolutionary significant.
- Go for the identification of duplication patterns in plant species that experienced WGD events before and after split and then relate your findings with the observed phylogeny.
good luck.
